# Comparison of Drug-Related Problems in COVID-19 and Non-COVID-19 Patients Provided by a German Telepharmacy Service for Rural Intensive Care Units

**DOI:** 10.3390/jcm12144739

**Published:** 2023-07-18

**Authors:** Joachim Andreas Koeck, Sandra Maria Dohmen, Gernot Marx, Albrecht Eisert

**Affiliations:** 1Pharmacy Department, Erlangen University Hospital, 91054 Erlangen, Germany; joachim.koeck@uk-erlangen.de; 2Department of Intensive Care and Intermediate Care, RWTH Aachen University Hospital, 52074 Aachen, Germany; sdohmen@ukaachen.de (S.M.D.); gmarx@ukaachen.de (G.M.); 3Hospital Pharmacy, RWTH Aachen University Hospital, 52074 Aachen, Germany; 4Institute of Clinical Pharmacology, RWTH Aachen University Hospital, 52074 Aachen, Germany

**Keywords:** telepharmacy, COVID-19, drug-related-problems, intensive care unit, pharmaceutical care

## Abstract

Telepharmacy is used to bridge the persisting shortage of specialist ward-based pharmacists, particularly in intensive care units (ICU). During the coronavirus disease 2019 (COVID-19), pharmacotherapy was rapidly developed, which resulted in multiple changes of guidelines. This potentially led to a differing risk for drug-related problems (DRPs) in ICUs. In this study, DRPs were detected in telepharmacy consultations of a German state-wide telemedicine network for adult patients in rural ICUs. The analysis included ICUs of ten general care hospitals with a total of 514 patients and 1056 consultations. The aim of this retrospective, observational cohort study was to compare and analyze the DRPs resulting from ICU patients with or without COVID-19. Furthermore, known risk groups for severe COVID-19 progression (organ insufficiency [kidney, liver], obesity, sex, and/or older age) were investigated with their non-COVID-19 counterparts. As a result, in both groups patients with acute renal insufficiency and without renal replacement therapy showed a significantly higher risk of being affected by one or more DRPs compared to patients with normal renal function. In COVID-19 patients, the initial recommendation of therapeutic anticoagulation (ATC-code B01AB ‘Heparin group’) resulted in significantly more DRPs compared to non-COVID-19 patients. Therefore, COVID-19 patients with therapeutic anticoagulation and all ICU patients with renal insufficiency should be prioritized for telepharmacy consultations.

## 1. Introduction

Telepharmacy was first mentioned twenty-four years ago to be “the provision of pharmaceutical care through the use of telecommunications and information technologies to patients at a distance” [1]. Since then, telepharmacy was in particular implemented for outpatients [2,3], but also for inpatients [4] and in intensive care units (ICUs) [5,6,7]. Telepharmacy services in ICUs are especially useful in settings where locally implemented ward-based pharmacists are rare. A current survey of chief consultants of 1549 German ICUs showed that only 58 of 168 respondents were supplied by ward-based pharmacy services [8]. This shortage has to be addressed by enhanced training and increased staffing of ward-based clinical pharmacists; meanwhile, telepharmacy services may bridge the gap. However, the extent of possible consultations is similarly limited for in-person and telepharmacy approaches; therefore, prioritization is useful, but effective risk assessment tools are still a matter of debate [9,10]. For a statewide telepharmacy service like ours, the selection of patient groups at the highest risk for DRPs or adverse drug events would help in case of excessive consultation inquiries and/or implementation into risk assessment tools. Telemedicine and telepharmacy have been able to maintain continuity of care for high-risk groups during the pandemic while maintaining social distancing and reducing the risk of infections [11].

The COVID-19 pandemic came up in late 2019. A pharmacotherapy was rapidly developed and initially found, e.g., in therapeutic anticoagulation [12,13,14] or immunosuppressants [15,16]. Our hypothesis was that these new drug treatments brought up a potentially differing risk for drug-related problems (DRPs) and, subsequently, adverse drug events in COVID-19 compared to non-COVID-19 patients in ICUs. Thus, the aim of the study at hand was to compare and analyze the DRPs resulting from the medication of COVID-19 patients compared with non-COVID-19 patients. Furthermore, known risk groups for severe COVID-19 progression (organ insufficiency [kidney and liver], obesity, sex, and/or older age) were investigated to their non-COVID-19 counterparts for their DRP risk.

## 2. Materials and Methods

### 2.1. Study Design, Setting, and Participants

The study followed the STROBE statement [17]. The telemedicine network was approved by the local Ethics Committee at the RWTH Aachen Faculty of Medicine (approval-No: EK 089/20). Telepharmacy consultations, including documentation and evaluation, are included in this ethics approval.

This is a retrospective, observational cohort study of telepharmacy consultations for adult patients in rural ICUs with/without confirmed COVID-19.

To date, the COVID-19 pandemic may be divided into six waves [18]; the first started in March 2020, and the sixth wave ended, according to the WHO, on 5 May 2023. For this analysis, patients were included that were consulted from March 2020 to August 2022 and when a current medication plan was retrieved.

Telepharmacy consultations are part of a German statewide telemedicine network (Virtuelles Krankenhaus VKh.NRW, North Rhine-Westphalia, Germany). Physicians in the ICUs of rural hospitals may register daily for a telemedicine consultation with a specialist intensive care physician of the RWTH Aachen University Hospital. The ten requesting general care hospitals with a size of 101–449 beds were connected to technical infrastructure and electronic health records (EHR). An EHR is used for documentation of patient data, the ‘Sequential Organ Failure Assessment’-score (SOFA) [19], and a short consultation report containing the physician’s and pharmacist’s recommendations. The physicians in the rural ICUs have access to these documents. The physicians in the rural ICUs selected adult patients for inclusion into the consultations that usually were followed until discharge from the ICU. Two times per week, a specialized pharmacist in intensive care medicine joined the interdisciplinary team to recommend drug therapy optimizations.

### 2.2. Variables

The pharmacist collected patient and health-related data, the current presentation, and data regarding pharmacotherapy. When applicable, admission diagnoses were classified using ICD-11 [20]. Anatomical therapeutic chemical (ATC) codes were taken from the “ATC/DDD Index 2022” [21].

A medication safety check included the analysis of indication, dose, dosing interval, therapeutic drug monitoring (TDM), drug application, drug therapy duration, contraindications, drug–drug interactions, potential dose adjustments, adverse reactions, and laboratory data [22]. All scheduled and on-demand medications were analyzed.

Drug-related problems (DRPs) were defined according to the Pharmaceutical Care Network Europe as “an event or circumstance involving drug therapy that actually or potentially interferes with desired health outcomes” [23]. In the case of detected DRPs, potential optimization strategies were discussed with the physician at the rural ICU. The pharmacist’s recommendations were documented in the EHR. DRPs, as discovered by the pharmacist, were categorized for analysis using the 27 categories of ADKA-DokuPIK classification [24,25]. A single DRP could be associated with multiple drugs.

### 2.3. Data Sources/Measurement

Drug indications were sought from nationwide or international guidelines and, when not available, from local protocols. Summaries of product characteristics (SmPC) were also included.

Dosing and dose adjustments were recommended based on the SmPCs. Depending on the clinical situation of the patient, evidence-based dosing recommendations may have differed from SmPCs. Dose adjustments for acute renal insufficiency were sought from DOSING (Heidelberg University Hospital, Heidelberg, Germany) and the Renal Drug Database (CRC Press, Abingdon, United Kingdom). For dose adjustments in case of liver cirrhosis, the website “Geneesmiddelen bij levercirrose” (https://www.geneesmiddelenbijlevercirrose.nl, accessed on 5 July 2023) was consulted. Dose adjustments for patients with obesity were sought from the book “Demystifying Drug Dosing in Obese Patients” [26].

The estimated glomerular filtration rate (eGFR) was calculated according to Levey et al. [27]. Drug selections and/or dose adjustments were recommended in acute renal insufficiency (i.e., eGFR of <60 mL/min/1.73 m^2^ and/or renal replacement therapy [RRT]). RRTs could include intermittent hemodialysis, prolonged intermittent kidney replacement therapy, continuous venovenous hemodialysis, or hemodiafiltration. Dosages were adjusted according to renal function and/or RRT.

A current serum creatinine value was queried in every consultation. When not available, serum creatinine range extracted from the SOFA-score was used for eGFR estimation (135/1056 consultations). In the case of a creatinine range of “<1.2 mg/dL”, an eGFR of >60 mL/min/1.73 m^2^ was estimated (*n* = 124). For the options “1.2–1.9 mg/dL” and “2.0–3.4 mg/dL”, the mean values (i.e., 1.5 mg/dL [*n* = 6] and 2.7 mg/dL [*n* = 4], respectively) were used. The option of “>5.0 mg/dL” was replaced by a serum creatinine value of 5.0 mg/dL (*n* = 1).

Drug–drug interactions were assessed using ID Pharma (ID Information und Dokumentation im Gesundheitswesen GmbH Co KGaA, Berlin, Germany), MediQ (Psychiatrische Dienste Aargau AG—mediQ Kompetenzzentrum für Medikamentensicherheit, Brugg, Switzerland), and/or Lexicomp Drug Interactions (UpToDate, Hudson, NY, USA).

For QTc-prolonging drugs, the Tisdale score was applied [28]. Incompatibilities were evaluated using Trissel’s™ 2 Clinical Pharmaceutics Database via IBM Micromedex^®^ (IBM, Armonk, NJ, USA) and the King^®^ Guide to Parenteral Admixtures^®^ Internet edition (King Guide Publications, Napa, CA, USA).

The body mass index (BMI) was calculated according to the World Health Organization (WHO), and weight classification was adopted, including the definition for obesity (BMI ≥ 30kg/m^2^) [29]. Drug dosing recommendations that differ from usual care were applied for patients with severe obesity (BMI ≥ 40kg/m^2^) [26,30].

### 2.4. Quantitative Variables and Statistical Methods

Numerical data were analyzed with mean, standard deviation, median, and/or interquartile range, whatever was applicable. Nominal scaled data were analyzed by the Chi-squared test when all single values showed *n* ≥ 5. In smaller *n*-values, Fisher’s Exact test was used. The Mann–Whitney U test was used for the analysis of differences in metric-scaled data. For small sample sizes in the Mann–Whitney U test (*n* ≤ 25), tabled exact critical values were used for larger sample sizes in a normal approximation. In all statistical procedures, a significance level of *p* = 0.05 was applied.

### 2.5. Special Drug Treatment

COVID-19-specific medications (i.e., antiviral and immunosuppressive therapies) were chosen due to the novel character of the disease. The current German and international guidelines were sought.

### 2.6. Special Patients on Risk for DRPs

Current guidelines list a special risk for severe COVID-19 complications among others for male patients, patients of older age, with renal insufficiency, liver cirrhosis, and/or (severe) obesity [31]. In addition, these patient groups, except for male patients, were seen as a marked risk for DRPs in the literature [32,33,34,35,36,37,38].

Specific DRPs were defined by the authors as those that only were mentioned because of acute renal insufficiency, liver cirrhosis, and/or obesity and would not have been stated without them.

## 3. Results

### 3.1. Participants

In total, 1139 consultations were performed; a current medication plan was retrieved for 1056 consultations (93%). So, 440 consultations for 191 COVID-19 patients (2.3 consultations per patient) and 616 for 323 non-COVID-19 patients (1.9 consultations per patient) were considered for this analysis (Figure 1).

In the same time period, the telemedicine network consulted 414 COVID-19 patients (3156 consultations). Thus, almost half of all COVID-19 patients were consulted by the clinical pharmacist at least once (46%, 191/414 patients).

### 3.2. Descriptive Data

COVID-19 patients included significantly more male patients compared to non-COVID-19 patients (68% versus 58%, Table 1, *p* = 0.022). In addition, COVID-19 patients showed significantly higher BMI (median 28 kg/m^2^ compared to 27 kg/m^2^ in non-COVID patients, *p* = 0.001). This result was underlined by significantly more obese COVID-19 patients (38% versus 29%, *p* = 0.037). Pre-existing liver and/or chronic kidney diseases were less common in COVID-19 patients versus non-COVID-19 patients.

### 3.3. Outcome Data

During the study, 707 DRPs were identified for 191 COVID-19 patients and 1077 DRPs for 323 non-COVID-19 patients. Both patient groups showed no statistically significant differences in the risk of suffering from one or more DRPs and in the DRPs/patient ratio. COVID-19 patients tended to show more patients with one or more DRP (90% versus 87%, Appendix A, *p* = 0.349). Furthermore, the DRPs/patient ratio was slightly higher for COVID-19 patients (3.7 vs. 3.3 DRPs/patient, Appendix A, *p* = 0.407). This increased ratio based on an increased consultations/patient ratio for COVID-19 patients (2.3 versus 1.9 consultations/patient, respectively, Figure 1).

### 3.4. Main Results

#### 3.4.1. DRP Categories and Associated Drug Classes

The five most frequent DRP categories comprised 62% and 54% of all DRPs in COVID-19 and non-COVID-19 patients, respectively (Figure 2, Appendix A, *p* < 0.001).

One significantly different DRP category between COVID-19 and non-COVID-19 patients was ‘TDM not performed or not considered’ (Figure 2, *p* = 0.044). Furthermore, significantly more DRPs associated with the ‘heparin group’ (ATC-code B01AB, e.g., heparin and enoxaparin) were found for COVID-19 compared to non-COVID-19 patients (46.3% vs. 15.1% of drug entries in this DRP category, *p* < 0.001). When focusing on drug entries (Figure 3, COVID-19: 774 drug entries associated with DRPs; non-COVID-19: 1217 drug entries), the only significantly different drug class between COVID-19 and non-COVID-19 patients was the ‘heparin group’ (*p* < 0.001). In the time span from November 2020 to October 2021, an increased thromboembolism prophylaxis dose was suggested for ICU patients in German COVID-19 guidelines [39,40]. In 37 cases of COVID-19 patients, this made anti-Xa monitoring necessary (e.g., because of decreased renal function); 34 DRPs (92%) resulted from the named time span.

#### 3.4.2. DRPs with COVID-19-Specific Medications

There were 26 COVID-19-specific DRPs in 25 patients (3.7% of all DRPs in COVID-19 patients, Appendix A). They addressed the drug classes ‘Glucocorticoids’ (*n* = 22), ‘Interleukin inhibitors’ (*n* = 3), and ‘Nucleosides and nucleotides excl. reverse transcriptase inhibitors’ (*n* = 1).

### 3.5. Other Analyses

#### 3.5.1. Patients on a Special Risk for DRPs: Age and Sex

In COVID-19 patients, different age groups (younger than 65 years of age versus 65 years and older) and sexes did not significantly affect the risk for one or more DRPs or DRP/patient ratios compared to non-COVID-19 patients.

#### 3.5.2. Patients on a Special Risk for DRPs: Acute Renal Insufficiency

Acute renal insufficiency was less common in COVID-19 patients compared to non-COVID-19 patients, although this failed statistical significance (46% versus 53% [87/191 COVID-19 patients versus 172/323 non-COVID-19 patients], *p* = 0.092).

COVID-19 and non-COVID-19 patients with acute renal insufficiency in one or more consultations had a significantly higher risk of showing at least one DRP compared to their counterparts (Figure 4, Appendix A, *p* = 0.004 and *p* < 0.001, respectively).

In addition, patients with acute renal insufficiency suffered from a significantly higher DRP/patient ratio compared to their counterparts (Figure 5, both *p* < 0.001).

Thirty-eight percent of COVID-19 patients and 34% of non-COVID-19 patients with acute renal insufficiency were affected by at least one DRP that was considered specific for renal insufficiency and would not have been mentioned without renal insufficiency (Appendix A, *p* = 0.564). Systemic anti-infectives were the most involved drug class in specific DRPs.

#### 3.5.3. Patients on a Special Risk for DRPs: Liver Cirrhosis

There were no significant differences detected between COVID-19 and non-COVID-19 patients with liver cirrhosis. During the consultations for rural ICUs, only 21 patients with liver cirrhosis of any degree were included; four COVID-19 patients and 17 non-COVID-19 patients. Twenty-six DRPs were considered specific for patients with liver cirrhosis; e.g., proton pump inhibitors or osmotically acting laxatives were involved.

#### 3.5.4. Patients on a Special Risk for DRPs: Obesity

COVID-19 and non-COVID-19 patients with obesity (BMI ≥ 30 kg/m^2^) were not observed to differ in their risk for suffering from one or more DRPs or in their DRPs/patient ratio.

Fifteen COVID-19 patients and 25 non-COVID-19 patients with severe obesity were included (BMI ≥ 40 kg/m^2^). They were affected by 25 DRPs that were considered specific for severe obesity. Mainly antithrombotic agents and systemic anti-infectives were involved.

## 4. Discussion

### 4.1. Key Results

To our knowledge, this is the first comparative study that investigated potentially differing DRP risks in COVID-19 compared to non-COVID-19 patients in ICUs that received telepharmacy consultations. Our initial hypothesis that there may be a differing DRP risk for ICU patients with and without COVID-19 could not be confirmed; in general, an equivalent risk for having one DRP and an equivalent DRP/patient ratio was found. However, this analysis revealed two major findings: First, patients with acute renal insufficiency (and without RRT) were generally identified as a risk group for DRPs, irrespectively of COVID-19 status. Second, besides other drug classes, COVID-19 patients suffered in particular from DRPs associated with the ‘heparin group’, which was targeted by telepharmacy consultations.

Patients in ICUs are at high risk for DRPs that may lead to adverse drug events. A recently published review found an incidence of adverse drug events of up to 96.5 per 1000 patient days [41]. Four years ago, a meta-analysis found reduced mortality when implementing a clinical pharmacist into the multidisciplinary ICU team [42]. This mortality reduction based on a significantly reduced rate of preventable adverse drug events. Unfortunately, training programs for ICU pharmacists, as implemented in the United Kingdom [43], are not yet in place in all countries. Furthermore, there is still a massive lack of ward-based ICU pharmacists, e.g., in Germany [8]. In this persisting shortage, ICU telepharmacy consultations might bridge this gap and enhance patient outcomes [5,44]. Telepharmacy was implemented during the COVID-19 pandemic, in particular for outpatients [2,45], also forced in Germany [46]. However, data from ICU populations is scarce [47]. Isleem et al. performed interviews with healthcare professionals to identify perceptions of pandemic ICU telepharmacy services. Healthcare professionals appreciated up-to-date information, particularly for COVID-19-specific medications or antithrombotics. In addition, they preferred ward-based consultations [48]. Telemedicine and telepharmacy can not replace in-person consultations, as physicians and pharmacists do not get into physical and emotional interaction with patients personally [11]. Locally derived training programs for ICU pharmacists have to be developed by responsible organizations in combination with increased ICU staffing with ward-based pharmacists.

The main topics of the analyzed telepharmacy consultations are generally in line with the existing literature. ICU patients with acute renal insufficiency (without RRT) were identified as a risk group for DRPs, irrespective of COVID-19 status. Patients with renal insufficiency were seen with a 16-times higher risk for adverse drug events compared to other ICU patients [49]. This highly corresponds to the results at hand. In addition, renal pharmacists have been implemented in different countries to account for the special risks for DRPs in this patient group [50,51].

Second, medication errors with antithrombotics in ICU patients are common and were described with a prevalence of 11 to 20% [52]. At the beginning of the COVID-19 pandemic, publications suggested an increased risk for venous thromboembolism and thus recommended therapeutic anticoagulation [12,13,14]. These recommendations were considered for ICU patients in German guidelines [39,40]. Our analysis confirmed that the extended use of therapeutic anticoagulation led COVID-19 patients to an increased risk for DRPs, although this could be attenuated by telepharmacy service. Antithrombotics are ranked among high-alert medications [53]. Thus, effective risk management programs are important, in particular for ICU patients. Risks are managed in some countries with the help of pharmacist-led anticoagulation services [54,55].

Generally, this analysis revealed that patients with acute renal insufficiency and on therapeutic anticoagulation were at marked risk for DRPs and should be prioritized in telepharmacy consultations and potential risk assessment tools (e.g., ‘dosage regimen’ [56]).

### 4.2. Strength and Limitations

The presented analysis method is useful for identifying relevant patient and medication groups that are at special risk for DRPs. On this basis, targeted methods for risk reduction may be developed to increase the quality of pharmacotherapy in ICUs.

The results are limited by the observational character of this study. Furthermore, there was no direct access to the local electronic patient record, and prescriptions were still performed paper-based. Therefore, disruption of information flow was inevitable and may have led to incomplete information at the level of consultations. DRP is an outcome parameter that is easily recorded, although the connection to potential patient harm might not be straightforward. Because of the small (sub-) group sizes, we did not conduct a multivariate analysis.

## 5. Conclusions

This is the first comparative study that investigated DRPs identified by a telepharmacy service for ICU patients with/without COVID-19. Two major findings revealed: First, patients with acute renal insufficiency and no renal replacement therapy were generally identified as a risk group for DRPs, irrespective of COVID-19 status. Second, COVID-19 patients were at an increased risk for DRPs with the ‘heparin group’ due to early COVID-19 guidelines that recommended therapeutic anticoagulation. These two patient groups should be considered for prioritization of telepharmacy consultations and potential risk assessment tools.

## Figures and Tables

**Figure 1 jcm-12-04739-f001:**
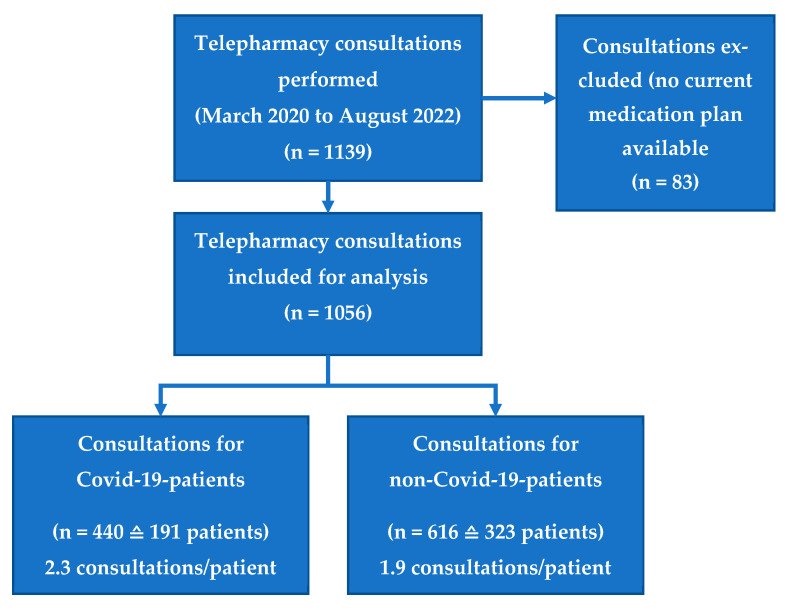
Patient flowchart.

**Figure 2 jcm-12-04739-f002:**
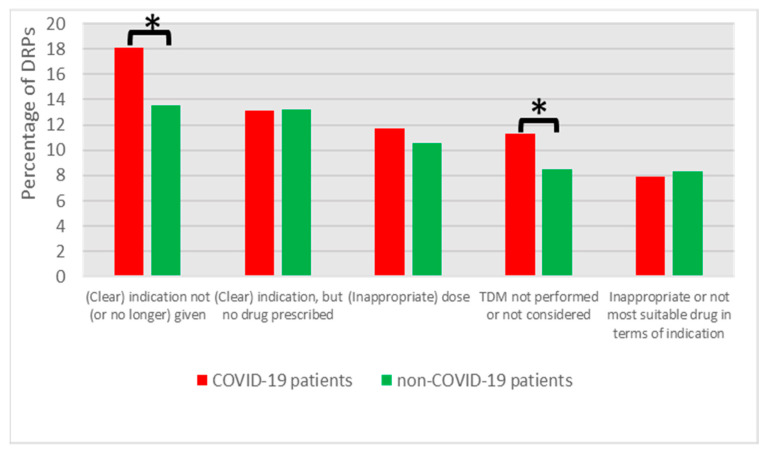
Top 5 DRP categories for COVID-19 and non-COVID-19 patients. Asterisk indicates significant difference (*p* < 0.05).

**Figure 3 jcm-12-04739-f003:**
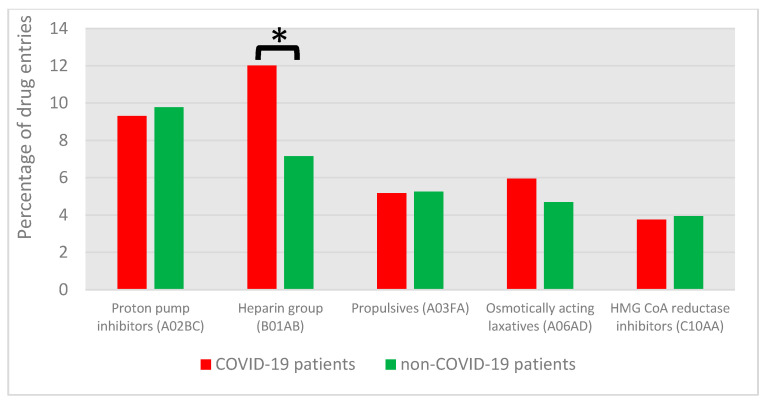
Top 5 ATC codes associated with DRPs for COVID-19 and non-COVID-19 patients. Asterisk indicates significant difference (*p* < 0.05).

**Figure 4 jcm-12-04739-f004:**
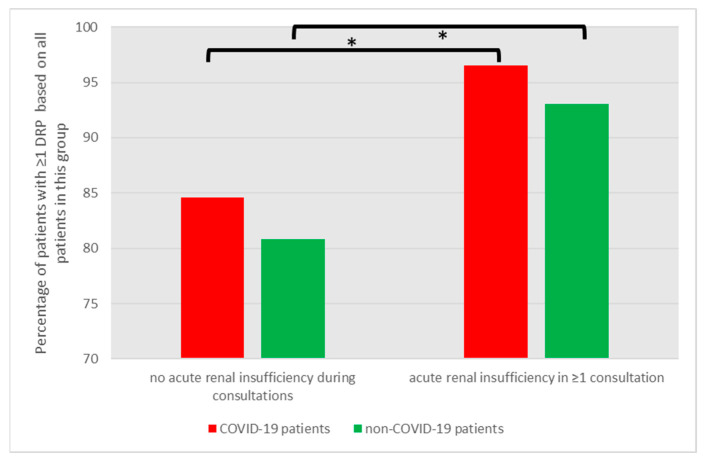
Inverse correlation of DRPs with renal function for COVID-19 and non-COVID-19 patients. Asterisk indicates significant difference (*p* < 0.05).

**Figure 5 jcm-12-04739-f005:**
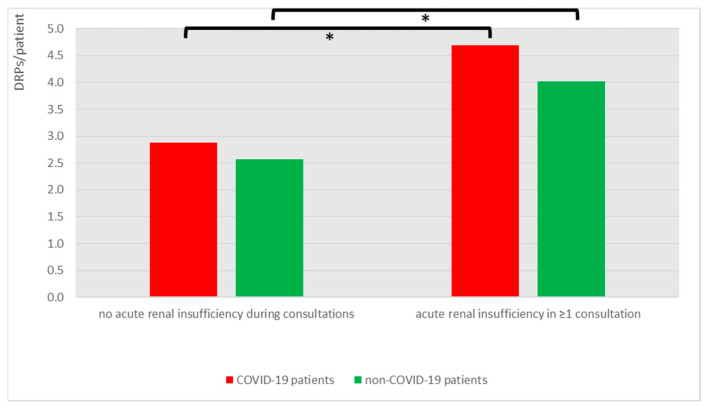
Significantly higher DRP/patient ratio in patients with acute renal insufficiency, independently from COVID-19 status. Asterisk indicates significant difference (*p* < 0.05).

**Table 1 jcm-12-04739-t001:** Patient characteristics.

	COVID-19 Patients (*n* = 191)	Non-COVID-19 Patients (*n* = 323)	*p* Value
**Age**			
Median (IQR)	66 (57–76)	70 (59–79)	
<65 years [*n* (%)]	88 (46)	121 (37)	0.055
65–74 years [*n* (%)]	47 (25)	84 (26)	0.725
75–84 years [*n* (%)]	47 (25)	93 (29)	0.303
>85 years [*n* (%)]	9 (5)	25 (8)	0.182
**Sex**			
Male [*n* (%)]	130 (68)	187 (58)	**0.022**
Female [*n* (%)]	61 (32)	136 (42)	**0.022**
**BMI**			
Median (IQR)	28 (25–31)	27 (24–31)	**0.001**
Underweight to preobese patients [*n* (%)]	119 (62)	230 (71)	**0.037**
Underweight (<18.5) [*n* (%)]	1 (1)	16 (5)	**0.007**
Normal range (18.5–24.9) [*n* (%)]	77 (40)	114 (35)	0.255
Preobese (25.0–29.9) [*n* (%)]	41 (21)	100 (31)	**0.020**
Obese patients [*n* (%)]	72 (38)	93 (29)	**0.037**
Obese class I (30.0–34.9) [*n* (%)]	39 (20)	52 (16)	0.215
Obese class II (35.0–39.9) [*n* (%)]	18 (9)	16 (5)	**0.049**
Obese class III (>40.0) [*n* (%)]	15 (8)	25 (8)	0.963
**Organ insufficiencies:**			
Patients with liver cirrhosis of chronic kidney disease of any type [*n* (%)]	30 (16)	73 (23)	0.059
Liver cirrhosis of any type [*n* (%)]	4 (2)	17 (5)	**0.040**
Chronic kidney disease of any type [*n* (%)]	26 (14)	59 (18)	0.170
**Time period of first telemedicine consultation**			
Period 1: first wave (2 March 2020–17 May 2020) [*n* (%)]	10 (5)	22 (7)	NA
Period 2: summer plateau (18 May 2020–27 September 2020) [*n* (%)]	6 (3)	83 (26)	NA
Period 3: second wave (28 September 2020–28 February 2021) [*n* (%)]	58 (30)	42 (13)	NA
Period 4: third wave (1 March 2021–13 June 2021) [*n* (%)]	45 (24)	28 (9)	NA
Period 5: summer plateau (14 June 2021–1 August 2021) [*n* (%)]	2 (1)	16 (5)	NA
Period 6: fourth wave (2 August 2021–31 December 2021) [*n* (%)]	43 (23)	54 (17)	NA
Period 7: fifth wave (1 January 2022–29 May 2022) [*n* (%)]	25 (13)	48 (15)	NA
Period 8: sixth wave (30 May 2022–31 August 2022) [*n* (%)]	2 (1)	30 (9)	NA

## Data Availability

Data are available upon request to the corresponding author.

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
