# Peer review of "Comparison of Drug-Related Problems in COVID-19 and Non-COVID-19 Patients Provided by a German Telepharmacy Service for Rural Intensive Care Units"

_jcm, 2023, doi:10.3390/jcm12144739_

Round 1

Reviewer 1 Report

1. The paragraph starting from Line 46 should be rewritten.

2. Line 62: COVID-19 is a disease so remove infection word, or use SARS-COV-2 infection. Also, correct "up to date" in the same line.

3. You cited figure in the text before table 1, but the table appeared first; change.

4. The subheading named 3.4 Main results. is misleading; change

5. Figure 2: the sentences below are overlapped especially for the first 3 columns and not clear for the readers.

6. The introduction and discussion need more details, particularly about the importance of telehealth/telepharmacy/telemedicine in the pharmacy field. Consider the following recent studies:

Alsayed, Ahmad R., et al. "Perspectives of the community in the developing countries toward telemedicine and pharmaceutical care during the COVID-19 pandemic." Pharmacy Practice 20.1 (2022): 1-12.     Khader, Heba, et al. " Pharmaceutical care and telemedicine during COVID-19: A cross-sectional study based on pharmacy students, pharmacists, and physicians in Jordan." Pharmacia 69.3 (2022): 891-901.   7. The tool used in this study is not free from limitations. TRP introduced 15 years ago and MPOP have more advantages in different aspects: AbuRuz, Salah M., Nailya R. Bulatova, and Almoatasem M. Yousef. "Validation of a comprehensive classification tool for treatment-related problems." Pharmacy World and Science 28 (2006): 222-232.   Alsayed, Ahmad R., et al. "Validation of an assessment, medical problem-oriented plan, and care plan tools for demonstrating the clinical pharmacist's activities." Saudi Pharmaceutical Journal 30.10 (2022): 1464-1472.

Some sentences are difficult to read.

Author Response

Please see also the corrected manuscript.

Please see also the corrected supplement.

Thank you for your valuable comments.

Reviewer 2 Report

I read with the interest the manuscript.

The manuscript presents the retrospective observational cohort study of telepharmacy consultations for adult patients in rural intensive care units (ICUs) during COVID 19 pandemic. The aim of the study was to compare drug related problems (DRPs)  in COVID-19- and non- COVID-19-patients in rural ICUs  in terms of DRP categories and involved drug classes. Its main contribution and strength are novelty and significance for implementation in future clinical work.

The manuscript is very relevant for the field but should be re-presented in several points.

Abstract should be reformulated as it is not understandable to read in isolation. Abstract should include the aim of the study,  type of analysis, participants and hospitals details.

Line 24 “ In COVID-19-patients, the ‘heparin group’ was significantly more involved in DRPs compared to non-COVID-19-patients, particularly the time where therapeutic anticoagulation was recommended”.. it is not clear to understand without reading the whole manuscript

Background and relevant research are briefly presented in the introduction. However, it would it be better to point out the authors’ aim(s) and hypotheses.

Line 53-54  “ To our best knowledge, nobody has investigated yet the risk for DRPs in COVID-19-compared to non-COVID-19-patients that were consulted by telepharmacy service forGerman rural ICUs”… The sentence in this form belongs to the Discussion and in deed it is mentioned there few times.. Rather state the aim and hypotheses simple and clear hear.

The methods are rather clearly presented and given in sufficient details to be reproducible.

Few details should be made clearer:

-          Line 58 “An ethics vote was obtained for the telemedicine network (EK 089/20).”  Please state it as clearly as stated under  Institutional Review Board Statement: “approved by the Ethics Committee of the University RWTH Aachen for the telemedicine network with the approval number EK 089/20.”

-          Line 60-61. “This is a retrospective observational cohort study of prospectively performed telepharmacy consultations”… as I understand this is retrospective, non- interventional, observational, cohort  study. “ Prospectively  performed”  is vague. Data are always prospectively collected. The question is when you started the study, before or after having collected data.

-          Line 62-63 “Up to dateg, the COVID-19 pandemic may be divided 62 into six waves [17]; the first started in March 2020, the sixth is still ongoing”- reformulate the sentence as the manuscript is reviewing now when the pandemic is declared to be over ( e.g. at the time the manuscript was written, the pandemic was still being).

-          Declare under 2. Materials and Methods; 2.1. Study design, setting, and participants: How many hospitals and which size hospitals were included in telemedicine network and this study?

In the Results part.. .Line 158 “ICUs of ten public general hospitals with a median of 248 beds” addresses this, but make it clearer in Material and Methods

Tables and figures are appropriate but not easy to read and follow. They are not readable in isolation. Firstly, I am missing all supplementary files (Figure S1, TableS1, Figure S2, Figure S3, Table S2, Table S3, Table S4,  Figure S4, Figure S5,Table S5, Figure S6, Table S6, Table S7). Are these supplementary files necessary at all? Legends of tables/figures are confusing, like written for poster. They should be more simple but clear. It is not necessary to repeat the results stated in the manuscript and then again in the legends. There is no consistency for all figures and tables.  If you state the results in the main text for the first table/figure , do the same in all other. Abbreviations in legends should be explained.

e.g. Line 188 and 200 ‘TDM not performed or not considered’…not clear, what is TDM?

The discussion is well written. The results are compared with the literature. However, as the authors missed to declare the aim and hypotheses in the Introduction, there is no reflection about reaching the aim and how the results fit with the initial hypotheses.

The conclusions are consistent with the evidence and arguments presented.

I have pointed out above my concerns about ethics statements.

Minor editing of English language required

Author Response

Thank you for your valuable comments.

Please see the corrected manuscript and supplement.

Round 2

Reviewer 1 Report

The mnuscript is improved and comments are addressed